# Electrospun Nanofiber Membranes for Air Filtration: A Review

**DOI:** 10.3390/nano12071077

**Published:** 2022-03-25

**Authors:** Yangjian Zhou, Yanan Liu, Mingxin Zhang, Zhangbin Feng, Deng-Guang Yu, Ke Wang

**Affiliations:** 1School of Materials and Chemistry, University of Shanghai for Science and Technology, Shanghai 200093, China; 192432632@st.usst.edu.cn (Y.Z.); yananliu@usst.edu.cn (Y.L.); 203613006@st.usst.edu.cn (M.Z.); 202342993@st.usst.edu.cn (Z.F.); 2Shanghai Engineering Technology Research Center for High-Performance Medical Device Materials, Shanghai 200093, China

**Keywords:** electrospinning, nanofiber, particulate matter, polymer, nanostructure

## Abstract

Nanomaterials for air filtration have been studied by researchers for decades. Owing to the advantages of high porosity, small pore size, and good connectivity, nanofiber membranes prepared by electrospinning technology have been considered as an outstanding air-filter candidate. To satisfy the requirements of material functionalization, electrospinning can provide a simple and efficient one-step process to fabricate the complex structures of functional nanofibers such as core–sheath structures, Janus structures, and other multilayered structures. Additionally, as a nanoparticle carrier, electrospun nanofibers can easily achieve antibacterial properties, flame-retardant properties, and the adsorption properties of volatile gases, etc. These simple and effective approaches have benefited from the significate development of electrospun nanofibers for air-filtration applications. In this review, the research progress on electrospun nanofibers as air filters in recent years is summarized. The fabrication methods, filtration performances, advantages, and disadvantages of single-polymer nanofibers, multipolymer composite nanofibers, and nanoparticle-doped hybrid nanofibers are investigated. Finally, the basic principles of air filtration are concluded upon and prospects for the application of complex-structured nanofibers in the field of air filtration are proposed.

## 1. Introduction

Air is one of the necessary conditions for human survival and life, covering every corner of the earth. Breathing is an unavoidable primary task in human life, and polluted air affects citizens more than other types of pollution because it is harder to avoid [1]. It is often said that water is the origin of life, but the status of air may be more significant. People can live for 5 days without water, but they can only live for less than 10 min without air—any longer, they will suffocate and die. People need about 1 kg to 2 kg of food and water for normal survival every day, and people need about 15 m^3^ of air for breathing a day, equivalent to a mass of 15 kg to 17 kg, which is nearly ten times that of water and food [2]. However, rapid economic development will inevitably bring about environmental damage, the continuous acceleration of industrialization has led to more and more particulate matter (PM) in the air, and the living environment of mankind is deteriorating. According to scientific research, air pollution can lead to many diseases, such as cardiovascular disease [3], malignant tumors, dry eyes, osteoporosis, fractures, conjunctivitis, infectious allergic diseases [4], inflammatory bowel disease, increased coagulation in blood vessels, a decreased glomerular filtration rate, and other physical diseases [5]. The serious harm caused by air pollution to the human body has also made air pollution control an urgent problem that needs to be solved. There is increasing concern about air pollution. Currently, it is listed as the most significant issue in the “2030 Agenda”. As one of the concepts of sustainable development, this is a global action plan developed by the United Nations [6]. Therefore, air pollution treatment has become a hot research topic. How to efficiently remove pollutants in the air is one of the difficult problems of air purification.

Air purification treatment is the best way to reduce air pollutants in the human body. The Chinese government has proposed several measures to combat smog, including reducing industrial and automobile emissions, improving the conversion ability of fuel and coal, and implementing artificial rainfall [7]. This pollution is mainly discharged outdoors, and the main living space of people is indoors. The threat of outdoor air to indoor air quality is imminent. Regarding personal physical protection, masks, air conditioning filters, and air purifiers are all common ways to deal with particle pollution. Regarding industrial production, various particulate pollutant capture methods have also been proposed [8]. Filtration can be divided into electrostatic dust collection filters and air filter filtration. The former mainly uses corona discharge technology to ionize the dusty airflow, giving the particles in the airflow a negative charge, and then captures and collects the particles on the dust collection substrate with a positive charge—it is an active filtering method. The latter uses specific materials to capture particles to achieve gas–solid separation [9,10]. Electrostatic dust collection has the advantages of no consumables, a long service life, and small airflow assistance, but its shortcomings are also obvious. In the case of high-voltage discharge, ozone will be generated, causing secondary pollution; the filtering effect of PM (such as PM2.5) is not very nice; and there are few places where it can be used.

Comparatively speaking, air filter membranes are now the research hotspot, as air filter membranes do not cause secondary harm, their filtration performance is controllable, and they have a wide application range. A perfect air filter material should have the following properties: (1) high filtration efficiency, (2) low air resistance, (3) nontoxic and does not cause secondary pollution. The core of filter-screen filtration is the selection and preparation of filter-screen materials. At present, the core functional layer of the filter screen responsible for removing PM is made of nonwoven fibers. The most widely used are melt-blown electret fibers and ultra-fine glass fibers [11]. With the continuous innovation and progress of science and technology, the requirements for air filters are becoming higher and higher. In traditional filter materials, the fiber diameter is usually on a micrometer scale, and the filter layer is usually composed of multiple layers. Although this filter can achieve high particle removal efficiency, it will increase the air resistance and increase the energy consumption of the equipment [12]. Moreover, the service life and dust-holding capacity of the filter membrane are gradually unable to meet the actual demand. Constantly updating the requirements promotes the progress of science and technology, and nanolevel fibers have begun to enter practical application research. Common methods for preparing nanofibers include electrospinning, template synthesis, interfacial polymerization, and self-assembly [13,14,15]. The electrospinning process has been widely used due to its simple and practical operation, and it is also a potential process for preparing air-filtration fiber membranes. The fiber membrane prepared by the electrospinning process has small fiber diameter, a large specific surface area, high porosity (up to 80% or higher), good connectivity, a small pore size, high surface roughness, and a low gram weight [16,17,18]. It can effectively solve the shortcomings of existing filters and more effectively remove the particle pollution in the air.

In order to understand the current research status of air-filter materials prepared by electrospinning, a search was conducted in “Web of Science” focusing on the themes of “air filter” and “electrospinning air filter”. The search outcomes are shown in Figure 1. The number of papers on topics related to “air filter” and “electrospinning air filter” has increased significantly every year. The literature on the subject of “air filter” has remained above 50,000 in the past four years, which shows that air pollution and its filtration and purification have attracted widespread attention. Searching for the subject of “electrospinning air filter “, the number of papers has increased in recent years, and the number of articles remains at more than one hundred, indicating that electrospun nanofiber membranes are becoming a new type of air filtration material with strong potential. This article introduces the research advances and the application potential of electrospun nanofiber membranes in the field of air filtration, and the advantages of electrospinning technology in air filtration materials.

## 2. PM and PM Filter Materials

### 2.1. PM

The formation of air pollution can be understood essentially as the inclusion of substances that cannot be metabolized in nature, such as toxic gases or suspended particles. The main sources of air pollutants can be divided into two types: (1) natural disasters (fog and haze caused by sandstorms, volcanic eruptions, fires, etc.)—these natural disasters are the main factors of air pollution in nature; (2) man-made causes (automobile emissions and industrial emissions)—man-made factors are the primary cause of air pollution [19]. The types of air pollution caused by human factors are numerous. From a spatial perspective, the types can be divided into point source, line source, and area source pollution. Point source pollution refers to related pollutant gases that are emitted from a designated point (commonly, factory chimneys, rural chimneys, power plants, etc.); usually, the polluted areas are smaller, but the concentration of the contaminant is high [20]. Line pollution refers to pollution discharged along a line, which is common in various transportation vehicles, such as vehicle emissions and running train track emissions. [21]. Surface pollution is relatively special. It refers to a number of small-scale pollution sources that accumulate over a long time to form serious pollution sources, usually formed by garbage incineration, agricultural incineration, cooking, etc.

Air is the most important source of supply for most organisms on the earth, and air quality directly affects human health [22]. The harm caused by particle pollution can be divided into two aspects:

(1) Harm to the natural environment. The hazards of PM to the natural ecological environment are divided into two types. One is that when PM exist in the air in high concentrations, it has a serious impact on solar lighting, the climate [23], air visibility, and ecosystems [24,25,26]. The second is the impact on agriculture. The suspension of PM affects the intensity of sunlight to a certain extent, leading to the weaker photosynthesis of crops. At the same time, because PM is partially deposited on the surface of crops, affecting their respiration, the combined effect of the two seriously affects the yield of crops [27].

(2) Harm to the human body. When PM reaches a higher concentration, the resulting pollution is the main factor causing human diseases [28,29,30]. PM is a trace air pollutant, which has been clearly recognized as a serious risk factor for premature death [31]. PM2.5 refers to the particles floating in the air whose diameter is less than 2.5 μm, which are small in size, strong in penetration, and easily carry various toxic substances that can cause cell death or organ dysfunction, which is extremely harmful [4,32,33,34]. PM2.5 enters the bronchi directly through the nasal cavity and then connects to the lungs, causing bronchitis, lung cancer, asthma, etc. [35,36]. When PM2.5 enters the human body’s circulation, such as the water circulation, it seriously affects the function of the kidneys and causes kidney failure in severe cases [37]. The destructive power that enters the human blood circulation is even greater. PM affects the transport of oxygen by hemoglobin. At the same time, the toxic substances carried by the PM can induce various cardiovascular diseases. The damage to pregnant women is especially serious, and the probability of abnormalities in infants and young children is increased [38,39].

### 2.2. Classification of PM Filter Materials

Filter materials are diverse because the requirements of filter materials are not the same in different environments; for example, when working in high-temperature surroundings, there is a need for high-temperature-resistant materials, and in high-humidity conditions, there is a need for hydrophobic air-filter materials. In terms of their development over time, air-filter materials can be roughly divided into the following two categories: (1) traditional filter materials, (2) modern filter materials. See Table 1 below for details.

#### 2.2.1. Traditional Filter Materials

Traditional air-filter materials are mainly particulate-filter materials and micron air-filter materials, such as bamboo charcoal bags and masks. Particle-filter materials are mainly used due to their low price, high-temperature resistance, and corrosion resistance, and they can achieve better results when used in some special environments. Micron-level and millimeter-level filter materials are now the mainstream; most of the materials in this area are used for masks and filters. However, these materials cannot meet the filtering requirements during their use, are easily blocked, and have a short working life. Refer to Table 1 for the specific advantages and disadvantages of different traditional materials.

#### 2.2.2. Modern Filter Materials

In comparison with traditional air-filter materials, modern air-filter materials, such as electrospun air-filter materials, have a high porosity, good pore connectivity, high removal efficiency, and low pressure drop [53,54]. In 2021, Cui et al. [55] used PVA-124 and TA electrospinning to prepare air-filtration fiber membranes. When the filtration efficiency of PVA-TA nanofiber membranes is set to 99.5%, the air passage resistance is 35 Pa. The membranes also performed well in terms of their overall performance. Modern filter materials are also relatively mature in terms of performance optimization, such as fiber filter membranes used in high-temperature conditions. Xie et al. [56] used electrospinning technology to prepare a PI-POSS@ZIF hybrid filter that achieved excellent filtration performance, with a high PM0.3 filter efficiency of 99.28% and an air passage resistance of 49.21 Pa at high-temperature conditions of 280 °C. A certain amount of ZIF was added during the preparation process for hybridization to improve the high-temperature resistance of the material. Common filtering particles include MOF, ZIF, and AgNPs.

## 3. Electrospinning

In air treatment, polymer membranes show great potential due to their excellent performance. At present, there are many technologies for preparing polymer membranes. Among them, electrospinning is a simple, versatile, and economical technology that can produce continuous nonwoven nanofibers from various polymer solutions [57,58]. Electrospinning is a “top-down” preparation technique in which the structure and size of the nanofibers can be customized using electrospinning parameters and polymer formulations [59]. Electrospinning has drawn attention due to its simple operation, low cost, and ability to produce good continuous fibers with diameters ranging from sub-micrometer to nanometer scale [60]. Therefore, polymer membranes prepared by electrospinning are also the preferred materials for air-filter media [61]. We need to conduct more in-depth research on electrospinning technology.

### 3.1. Principles and Equipment of Electrospinning

Electrospinning is a method of producing fibers by stretching a polymer by electric field force [48]. It uses the interaction between the medium and the electric field. The preparation of nanofibers by the “top-down” method is a simple, efficient, economical, and flexible method. As shown in Figure 2, four key devices are needed to achieve this process: (1) a high-voltage generator; (2) an infusion pump, which can accurately control the speed of the distribution of the solution or melt; (3) a metal needle or spinneret; and (4) a collector [62]. After the resurgence of electrospinning in 1995, it was considered as a process of continuous splitting in the atomization region to produce nanofibers. After ten years, the unstable atomization region was recorded as a high-frequency jet and curved whipping region. On this basis, the extremely fast stretching/drying process of the working liquid in the electrospinning process was divided into three steps (Taylor cone, linear jet, and unstable region) [63]. The electrospinning process has three stages. (1) the polymer solution in the syringe flows out of the spinneret under the action of the injection pump and forms spherical droplets under the action of gravity, thrust force, and the surface tension of the droplets. (2) At this time, the high-pressure generator is connected, and the surface of the droplet starts to accumulate charges. Under the action of the electric field force, the droplet stretches from a spherical shape to a conical shape. At this point, the droplet surface is subjected to the joint action of the gravitational Coulomb force and surface tension. When the Coulomb force is greater than the surface tension, the formed cone (Taylor cone) emits a straight jet [64]. (3) In the third stage, the linear jet shows instability, which is due to the rapid evaporation of the menstruum of the jet, and the interaction of the positive charge and the electric field on the surface creates whipping and bending, forming polymer fibers. 

Electrospinning is a fiber-processing technology with a simple operation but great potential. The diversity of the fiber structures and the adjustable preparation design mean that it is widely used. The advantages can be summarized as the simple experimental conditions, the low experimental cost, the high fiber yield, and the high applicability of the materials [65].

### 3.2. Influencing Factors

The influencing factors that affect the electrospinning process are divided into three types: system parameters, environmental parameters, and process parameters [66]. As can be seen in Figure 3, the three parameters each contain a variety of different influencing factors, involving the properties of the polymer itself, the environmental impact, and the influence of the equipment and process. By changing the process parameters, such as the electric potential, the flow rate, the polymer concentration, and the distance between the electrodes, a variety of electrospinning products can be obtained, including dense and hollow ones for specific applications and porous structures [66]. The following subsections will discuss the three aspects of the system parameters, the process parameters, and the environmental parameters.

#### 3.2.1. System Parameters

As shown in Figure 3, the system parameters contain the solvent, solution viscosity, surface tension, dielectric constant, electrical solution concentration, and solute molecular weight. These influencing factors do not exist in isolation. For example, in the case of the solution viscosity, the concentration is proportional to the viscosity—the higher the concentration, the higher the viscosity of the solution. The polymer molecular weight also affects the solution viscosity—the polymer molecular weight is generally proportional to the solution viscosity [67]. When other factors are consistent, the viscosity of the solution increases to a certain extent, which is conducive to the preparation of nanofibers with good morphology. However, if the viscosity of the solution is too high or too low, it increases the probability of the formation of beaded fibers [68]. The higher the molecular weight of the polymer, the higher the probability of forming linear fibers, which is partly due to the influence of the molecular weight of the polymer on the viscosity of the solution. The solvent is a significant factor that affects the electrospinning process. The solubility of the solvent for a certain polymer and the volatility of the solvent are two important selection criteria [69]. The adjustment of the solvent can be divided into two aspects: the selection of a single solvent and the adjustment of the ratio of mixed solvents. The selection of single solvents mainly takes into account the solubility of polymers in solvents, while mixed solvents need to consider the ratio of multiple solvents and the effects that mixed solvents can achieve. For example, two solvents with different volatilities can make use of their different evaporation rates to prepare porous structures.

When other factors are not changed, the electrical conductivity also has a greater impact on the formation of fibers. As the electrical conductivity increases, the fiber diameter decreases, and the prepared fiber beads are greatly reduced. However, too high an electrical conductivity causes serious discharge, reduces the uniformity of the fiber diameter distribution, and results in the failure of the electrospinning. Topuz et al. [70] found that after adding a certain amount of ammonium salt into the solution to improve the conductivity of the solution, the solution would undergo a spinning process at a lower concentration. The dielectric constant has a similar effect on the electrospinning process. The dielectric constant mainly refers to the nature of the solvent. Within a certain range, the larger the dielectric constant of the same solvent, the smaller the nanofiber diameter; too high a dielectric constant affects the jet stability.

#### 3.2.2. Process Parameters

The process parameters are related to the electrospinning equipment, including the flow rate, receiver distance, voltage, and spinneret diameter.

In the process of electrospinning, the driving force of solvent evaporation and polymer filament formation is the electric field force, and the electric field force is controlled by the voltage. It is found that any electrostatic spinning process requires a minimum starting voltage [71]. In Taylor’s experiment, it was found that when the sum of the surface tension and the gravity of the droplet at the end of the spinneret was less than the sum of the electric field force and the thrust of the syringe pump, the tip of the droplet produced a straight jet. In the study, because the gravity and thrust of the droplet were very small, it could be considered that when the force of the electric field on the droplet was equal to the surface tension, a linear jet produced [72]. It can be concluded that when the voltage is greater than the minimum pressure, the higher the voltage will be, to a specific extent; the faster the solvent will evaporate; the faster the fiber will become filaments; and the thinner the fibers will be. When a certain limit is exceeded, the average diameter does not decrease but increases. For example, in the experiment of Seyed et al. [73], when the voltage reached 28 kV, the average diameter of the prepared glass fibers increased. This may be because the voltage was too high, and after forming a straight jet, the polymer solidified too fast and had not been stretched in the electric field.

The effect of the flow rate is mainly on the fiber diameter and fiber uniformity. A small flow rate can produce finer fibers. A rise in the flow rate increases the diameter of the fiber, but when the flow rate is too high, this results in the formation of a beaded morphology [74]. The receiver distance is the distance from the end of the spinneret to the receiving device. Within a certain acceptance range, the farther away the receiver is, the greater the benefit to the volatilization of the polymer–solvent and the stretching of the electric field force. This forms fibers with better surface morphology and a smaller diameter. Regarding the influence of the spinneret diameter on the fiber, there may be deviations in common sense. It is believed that the spinneret diameter affects the fiber diameter. In the research of He et al. [75], it was found that the solution viscosity of the spinneret with a larger diameter increased at the end of the spinneret, which led to the increase in the fiber diameter. Therefore, the increase in the fiber diameter caused by the increase in the spinneret diameter was essentially due to the increase in the viscosity. 

#### 3.2.3. Environmental Parameters

The influence of environmental factors on the electrospinning process is not as great as the previous two factors, which is mainly because both the temperature and humidity can be controlled by instruments. The influence of temperature is closely related to the influence of other parameters; for example, with an increase in the temperature, there is an increase in the solvent volatilization rate, the polymer solidification is faster, and there is a decrease in the fiber diameter. This is reflected in the studies of Cui et al. [76] and Huang et al. [77]. The influence of humidity is mainly reflected in the volatilization rate of the solvent. As the humidity increases, the volatilization rate of the solvent decreases to form a beaded fiber. For some water-absorbing polymers, the influence of humidity is very important, such as polyvinylpyrrolidone (PVP). Therefore, the adjustment of environmental parameters requires the integration of polymer properties and solvent properties, and the impact of environmental factors on different electrospinning fluid systems is quite different.

#### 3.2.4. Other Influencing Parameters

Besides the properties of raw materials and the influencing factors in the preparation process, the post-treatment of products is also an important factor affecting the performance. For example, the heat treatment of products can improve the mechanical properties and stability of materials. In the face of industrial-waste-gas emission at temperatures as high as 260 °C, it is urgent to prepare refined and high-temperature resistant filter materials. Electrospun fibers prepared at room temperature can easily be oxidized at high temperatures, and the filtration efficiency is reduced. By designing heat-treatment conditions, the structure of electrospun fiber membranes was improved to achieve the advantages of a small size effect, an adjusted internal connectivity, improved stability, and enhanced mechanical properties. Chen et al. [78] prepared electrospun nanofibers by blending PA66 and PVB. Through SEM images, it was obvious that the fiber diameter decreased and the arrangement of the fiber structures was more regular. Compared with untreated nanofibers, the filtration efficiency of W-15 G nanofiber membranes for 0.3 μm airborne particles was as high as 99%. The cross-link agent was added in this composite fiber, and the rapid cross-linking of the composite fiber was realized through the heat-treatment process. This method can significantly improve the solvent resistance and mechanical properties of nanofibers. Gui et al. [79] annealed at 170 °C by adding a certain amount of citric acid into the working solution of the electrospinning precursor. When the annealing temperature was 160 °C, the tensile strength of the CS/PEO/CA nanofiber felt was as high as 8.11 Mpa. In addition, for the nanoparticle–nanofiber mixture, heat treatment could not only reduce the fiber diameter but also improve the crystallinity of the nanocrystals. Zhang et al. [80] prepared composite fiber membranes by electrospinning water-soluble PVA/CNCs. The existence of CNCs significantly promoted the growth of PVA crystals under heat treatment. Zhang et al. [81] heat-treated PAN/PMIA composite fiber membranes at 140~220 °C for 30 min. There was no entanglement between the fibers, and the breaking strength of the composite fiber film decreased slightly, while the elongation at the break increased significantly. After heat treatment, the filtration efficiency of PAN/PMIA composite fiber membranes remained above 99%. In summary, heat treatment, as one of the post-treatment methods for materials, can improve the performance of nanofibers after preparation, such as the fiber stability and the mechanical properties. Therefore, the reasonable post-treatment of products is also one of the ways to improve materials’ properties.

### 3.3. Types of Electrospinning Process

With the progression of time, the electrospinning process has gradually advanced. Electrospinning technology can be divided into single-fluid electrospinning technology and multifluid electrospinning technology, according to the amount of electrospinning solution used. The single-axis electrospinning process has higher requirements for the spinnability of the materials, and it is also subject to greater restrictions in practical applications. Fluid electrospinning technology can be divided into coaxial electrospinning, side-by-side electrospinning, and multistage electrospinning technologies. The fibers prepared by coaxial electrospinning have a core–shell structure [82], as shown in Figure 4f. The nanofibers prepared by side-by-side electrospinning have a Janus structure [82], as shown in Figure 4g.

#### 3.3.1. Single-Fluid Electrospinning

Single-fluid electrospinning is the most basic process. As shown in Figure 4a,e, single-fluid electrospinning only involves one fluid, and the process equipment is relatively simple. All the derived processes of multifluid electrospinning, needle-free electrospinning, and molten electrospinning are based on the single-fluid electrospinning process. Single-fluid electrospinning has greater restrictions on the materials, and the polymer to be prepared is required to have spinnable properties. At present, there are only just over a hundred kinds of spinnable polymer, which greatly limits the development of electrospinning. In addition to preparing nanofibers from a single polymer in the conventional sense, it can also be used for the blending of multiple polymers or the blending of polymers and dopants [83]. In addition, there is also emulsification electrospinning, where the hydrophilic substance is dispersed in the water phase and then dispersed into the oily substance before electrospinning is performed.

#### 3.3.2. Multifluid Electrospinning

(1) Coaxial electrospinning

There are plenty of limitations in the application of uniaxial electrospinning, such as the limited spinnable materials and the single performance. With the continuous innovation of researchers, multifluid electrospinning has been developed to solve these problems. As early as 2002, Loscertales et al. [84] used an electro-hydraulic dynamic (EHD) force to generate a coaxial jet of immiscible liquid with a diameter in the nanometer range to prepare nanofibers with a core–shell structure. This experiment also opened a new chapter for the development of electrospinning technology. Coaxial electrospinning uses a special spinning head, as shown in the concentric circle structure in Figure 4b,f. The two needles are arranged into a core (center needle) and a shell (annulus), and different solvents can be added to the core layer and the sheath layer to achieve the purpose of preparing composite materials. When performing coaxial electrospinning, different electrospinning solutions can be used for the core layer and sheath layer, and the sequence of the core and sheath can be changed to produce a variety of composite materials [85,86,87]. Fiber structure types prepared by coaxial electrospinning include flat ribbon fibers [88] and hollow fibers [89]. The fiber thickness and smoothness can be changed by using coaxial electrospinning. Coaxial electrospinning greatly broadens the application prospects of electrospinning technology and overcomes many problems existing in uniaxial electrospinning. In coaxial electrospinning, the flow of the core liquid and sheath liquid is not completely stable (e.g., instability caused by bending and whipping) and the core liquid may deviate from the center area of the sheath liquid [90]. In extreme cases, the core may be exposed to the environment because it is not completely covered by the sheath [91]. The properties of the two fluids are the key to the preparation of smooth fibers by coaxial electrospinning [92]. Researchers believe that there are forces between the core fluid and the sheath fluid, such as friction and viscous drag, resulting in shear forces between the fluids. When the shear force generated by the viscosity of the sheath fluid is greater than the tension between the core–sheath fluid, the core–sheath fiber can be prepared stably [93].

In the early stages of the development of the coaxial electrospinning process, researchers believed that both the core fluid and the sheath fluid should be spinnable during coaxial spinning. However, in 2010, Yu et al. conducted a series of experiments that disproved this traditional understanding [94] and, for the first time, used a nonspinning solvent as the core solution and a spinnable solution as the sheath to successfully prepare nanofibers. This breakthrough development also brought new vitality to coaxial electrospinning technology, and the problem of material selection was solved. The application of coaxial electrospinning in the experiment could not only prepare core–sheath structure nanofibers, but could also prepare finer uniaxial nanofibers. In the experiment, the use of a pure solvent in the sheath liquid could make the spinning process more stable and reduce the flow cut-off, resulting in a finer and smoother fiber diameter. Meanwhile, the blockage of the spinneret plate could be effectively reduced, and the negative influence brought by the external environment was reduced [95]. The emergence of coaxial electrospinning has largely solved the problem of polymer spinnability, and the superiority of electrospinning technology has also been brought into play. Nowadays, coaxial electrospinning is also booming. For example, Lv et al. [96] used modified coaxial electrospinning technology to prepare core–sheath nanostructure fibers for drug release.

(2) Side-by-side electrospinning

The working principle of side-by-side electrospinning is similar to that of coaxial electrospinning. Side-by-side electrospinning is a kind of multifluid electrospinning technology. To prepare nanofibers with a Janus structure, two different solutions are needed, as shown in Figure 4b,g. The biggest feature of the prepared fiber structure is that the fiber has a Janus structure, and both materials can directly contact the environment. Side-by-side electrospinning technology is one of the most difficult challenges in the field of electrospinning technology [97], and there are currently not many studies on this topic.

As it is traditionally conceived, the side-by-side electrospinning process requires two parallel spinning heads to form a side-by-side spinning plate, as shown in the last case in Figure 4b. However, since the same charge is applied to the two fluids, and due to the principle of like–like repulsion, the two materials are often separated due to repulsion, which poses great operational difficulties. After the researchers’ exploration and innovation, Yu et al. [97,98] proposed methods to improve the spinneret, such as arranging internal capillaries in a circular spinning head to realize side-by-side electrospinning or using an eccentric spinning head that has an elliptical outer nozzle, including an inner circle side and a crescent side. As shown in the eccentric circle in Figure 4b,d, the spinning head is nested, but the center of the circle is not on the same axis. This can be called an eccentric spinning head. The prepared fiber has a Janus structure. From the point of view of equipment, the difference between side-by-side electrospinning and coaxial electrospinning is not large, and fibers with completely different structures can be obtained by changing the structure of the spinneret. 

In terms of the experimental conditions, side-by-side electrospinning is stricter than coaxial electrospinning. Usually, the side-by-side electrospinning process can spin normally when the two fluids have good compatibility and viscosity, which greatly limits the popularization of the side-by-side electrospinning process [99]. With the continuous innovation of technology, improvement methods for side-by-side electrospinning have been put forward. For example, in side-by-side electrospinning, a layer of pure solvent is wrapped outside the two electrospinning solutions, which can greatly improve the filament-forming conditions. Research into side-by-side electrospinning is also going on continuously. For example, Li et al. [100] used PVP K90 and EC as a polymer matrix and loaded KET and MB to prepare Janus fibers for controlled drug release. 

(3) Triaxial electrospinning

The demand for nanomaterials with complex structures has promoted the development of multifluid electrospinning processes [101]. There are many designs of three-axis electrospinning spinnerets, as shown in Figure 4c. Zhao et al. [102] used modified triaxial electrospinning to prepare a series of functional nanoparticles and nanofiber membranes for the natural degradation of antibiotics. In 2016, Yu et al. [103] used a modified triaxial electrospinning process to prepare core–shell polymer-PL nanocomposites for oral colon-targeted drug delivery, as shown in Figure 5. Then, Yu et al. [104] adopted the sustained-release core–shell fiber prepared by the modified triaxial electrostatic spinning method for the bidirectional release of drugs, and the triaxial materials provided more possibilities for the fiber to carry drugs. In 2020, Wang et al. [105] adopted a new strategy of triaxial electrospinning technology to prepare three-layer nanolibraries, that is, to build drug reservoirs in core–shell nanofibers.

It can be deduced from the fact that electrospinning technology can prepare high-quality nanostructured fibers. In the near future, the technology could be widely used in various fields. The most important problem of electrospinning is how to reasonably combine the working fluid and electric energy [86,106]. The spinneret is the key to the improvement of the electrospinning process and also the focus of the innovative process. The application of single-fluid electrospinning is limited by the materials. Through the multifluid electrospinning process, more diversified nanostructured fibers can be prepared, which can be used in such fields as controlled drug release [96,107,108,109,110], wound dressing [111], food packaging [112,113,114,115], antibacterial materials [116,117], tissue engineering [118], and water treatment [119,120]. However, due to the limitation of polymer spinnability in the early stage, there is little research on the preparation of air-filtration fiber membranes by the multifluid electrospinning process. Multifluid electrospinning solves the materials limitation and can prepare air-filtration fiber membranes with various structures and excellent performance, which has great potential and research value in the field of air purification.

## 4. Electrospun Fibrous Membranes for Particle Filtration

PM has different manifestations in different scenarios and also needs different filtration methods. For example, factories operating under high-temperature conditions need high-temperature-resistant materials, and normally, due to smog and other conditions, people need masks to travel and filter screens are needed for automobile exhaust treatment. In each of these scenarios, the demand for materials is different. The research on fiber materials is a hotspot now, and most scenarios need to be solved by modifying the properties of the fiber material. Nanofibers prepared by electrospinning technology have excellent properties, including a large specific surface area, high porosity, uniform fiber diameter distribution, and excellent surface adhesion, which are significant for air filters [121,122]. The performance of the air-filter element is closely related to the structural factors of the nanofiber membrane. As mentioned above, the filtration performance and fiber diameter, specific surface area, and fiber thickness are closely related to the filtration efficiency and pressure drop of the fiber membrane.

### 4.1. Filtration Principle

#### 4.1.1. Single-Fiber Filtration Principle

Air-filtration processes can be divided into two kinds, in theory: the steady-state filtration process and the unsteady-state filtration process [123]. In the steady-state filtration process, as the name implies, the performance of the membrane does not change during air-filtration process, and so the filtration efficiency and pressure drop do not change with time. This is solely dependent on the intrinsic qualities, particle properties, and air velocity of the filter medium. For unsteady-state filtration, the filtration efficiency and flow resistance change with time due to the accumulation of particles on the filter [123,124]. In the 1930s, Freundlich summarized the rules of filtration through research on aerogel filtration and then proposed an air-filtration model based on Brownian motion. In 1936, Kaufmann expanded the idea of air filtration by one step, thanks to advances in theoretical science and technology. He combined Brownian motion and inertial deposition to summarize the mathematical model of fiber filtration [117]. Then, the theory advanced to single-fiber filtration theory, which integrated Brownian motion, inertial deposition, and the interception effect mechanism [123]. In the subsequent theoretical development, it was found that, besides the above reasons, the gravity effect and electrostatic effect have a significant influence on particle filtration. Therefore, five mechanisms are mainly used for fiber filtration to capture particles, as shown in Figure 6a: the interception effect, inertial impaction, diffusion effect, gravity effect, and electrostatic effect [125,126,127].

(1) Interception effect

The fiber film formed by stacking fibers is not regular in its fiber arrangement, and the motion track of fine particles in the airflow around the fibers is streamlined [11,125]. As shown in Figure 6b, if the center of each aerosol particle follows a streamline and does not deviate from the streamline unless there another external force, the pollutant particles do not come into contact with the surface of the fiber filter material [128]. As the gas flow is subjected to van der Waals forces, the fine particles come into contact with the surface of the fibrous filter material to intercept the PM—this is the interception effect [129].

(2) Inertial impact

The stacking of fibers in the fiber membrane is not regular and the arrangement is very complicated [11]. The complex arrangement of the fibers causes reverse, offset, and circulation effects when the airflow flows over the surface of the fibers. When the airflow is not flowing along the streamline, the particles in the air are separated from the airflow due to inertia and then deposited on the surface of the fiber membrane. The effect of inertial deposition is evident when facing particles larger than 0.3–1 μm. This mechanism has become the primary trapping mechanism, especially for larger particles and higher airflow velocities [121,130]. The greater the mass of the PM particles, the faster the airflow, and the more obvious the effect of inertia.

(3) Brownian diffusion

The phenomenon whereby suspended particles never stop making irregular movements is called Brownian motion. The diameter of aerosol particles in the air is very small, and Brownian diffusion is the main form of diffusion. Brownian motion is also an important mechanism in the air-filtration process, and random Brownian motion may also cause particles to collide with fibers, resulting in deposition [121]. Aerosol particles deviate from the original streamline under the action of Brownian diffusion. The effect of Brownian motion is more significant for particles smaller than 0.1 µm in size, resulting in diffusion and deposition around the fiber surface [131,132].

(4) Gravity effect

Aerogels are affected by gravity during their movement, but the particles are too small to be affected by gravity. The settling effect is entirely negligible when the particles are smaller than 0.5 m [133].

(5) Electrostatic effect

The electrostatic effect is a common phenomenon in life. For example, computer screens easily accumulate dust. In air filtration, if there is a charge between the particle and the fiber membrane, the particle’s orbit changes due to the electrostatic effect; the result is to attract particles deposited on the surface of the fibers [134,135]. At the same time, electrostatic adsorption can make particles adhere more firmly to the surface of the fibers. The electrostatic effect is widely used in particle-filter equipment, as electrostatic force can greatly improve the degree of filtration of PM [136].

In the actual fiber filtration process, different filtration mechanisms are specific for contaminants with different particle sizes. The five filtration mechanisms complement each other, combining interception, inertial collision, Brownian diffusion, static electricity, and gravity. In addition to the above mechanisms, the Van der Waals force also has a significant effect on the filtration efficiency. For example, Chen et al. [137] pointed out that the Van der Waals force and coulomb force had a very significant effect on the filtration of 100 nm particles.

#### 4.1.2. Evaluation of Filter Material Performances

We mainly discuss the steady-state filter system. The unsteady state can be analyzed by using the infinitesimal method to decompose it into a steady state. Fine fibers, dense aggregation structures, thickness, the electrostatic effect, and other factors can all help to intercept particles in the air completely. The performance of the particle filter is judged by the following three indexes: (1) filtration efficiency, (2) airflow resistance, and (3) quality factor [138]. The rating of these three indexes is the standard that evaluates air-filter materials. 

(1) Filtration efficiency

The filtration efficiency of an air-filtration material refers to the ratio of the number of particles filtered by the filtration fiber membrane to the initial number of particles in the gas as the air with the contaminated particles passes through the filtration fiber membrane. The filtration efficiency is a critical metric for air-filtration materials, as it decides whether they are suitable for practical use. There are a variety of filter efficiency tests available, including the oil mist test, the weight method, the sodium flame method, the colorimetric method, the fluorescence method, the photometer scanning method, and the counting method. The counting approach is the most widely utilized method. The testing of filtration efficiency mainly measures the difference in the number of particles on both sides of the filter membrane. Figure 6c depicts a schematic diagram showing filtration efficiency. The purpose of testing filtration efficiency is achieved by calculating the difference in the number of particles in the gases flowing through the flow meter.

In the actual test, the collection efficiency of the fiber filter medium of the fiber membrane is calculated as follows:(1)$$\eta =\left[1-\frac{C_{2}}{C_{1}}\right]\times 100\%,$$
where *C*_1_ is the number of examples detected by flowmeter 1, *C*_2_ is the number of examples detected by flowmeter 2, and *η* is the filtration efficiency of the fiber membrane.

The filtration efficiency of the fiber membrane can be obtained using Formula (1) in the real test, but the relationship between the filtration efficiency and the structure of the fiber membrane cannot be observed in the theoretical analysis. It has been found that the filtration efficiency of the fibers can be predicted based on the well-known Kuwabara model to find the overall particle filtration efficiency (*η*) of the nanofiber membrane, and the formula is as follows:(2)$$\eta =1-\exp\left[\frac{-4\eta_{s}\alpha L}{\pi d_{f}\left(1-\alpha \right)}\right],$$
where *η_s_* is the filtration efficiency of the individual fibers, *α* represents the fiber volume fraction, *d_f_* represents the average fiber diameter, and *L* represents the thickness [121,125,139].

It can be seen from the Kuwabara model that the filtration efficiency of the fiber membrane is directly related to the filtration efficiency, volume fraction, fiber diameter, and thickness of a single fiber. The smaller the diameter, the greater the thickness, and a higher filtration efficiency for a single fiber is beneficial to the filtration efficiency of the fiber membrane. In this model, the effect of electrostatic force on the filtration efficiency is not considered. When the air-filtration fiber membrane is prepared by the electrostatic spinning process, the fiber membrane generates electrostatic force due to high pressure, which is beneficial to the adsorption of particles in the air. However, electrostatic force is gradually lost after the film preparation, and this electrostatic force, as well as other measures of voltage, are needed to assist in the adsorption, so the electrostatic force adsorption is not discussed here.

(2) Pressure drop

The second factor that affects the performance of fiber membrane filtration is the pressure drop, which characterizes the air resistance. The pressure drop refers to the fact that when particles in the air pass through or collide with the fiber membrane, the pressure of the fiber membrane on both sides of the membrane is different, due to the barrier of the membrane. Generally speaking, the pressure decreases after filtration, which is also known as the pressure drop. The decrease in the nanofiber membrane pressure is due to the large gaps between the nanofibers, that is, the high porosity. When the diameter of the nanofibers is comparable to the average free path of the air molecules (66 nm under normal conditions), the gas velocity at the fiber surface is not zero, due to the “slip” effect. Due to the “slip” effect, the resistance from the nanofibers toward the airflow is greatly reduced, thereby greatly reducing the pressure drop [140].

Nanofiber membranes have a small diameter and high porosity, which can promote the flow of gas on the membrane and thus play a role in reducing the pressure drop. A low thickness, porosity, and a low fiber diameter allow air to pass through under reduced pressure. The pressure drop (ΔP) is related to the drag coefficient and can be expressed by the following mathematical model [141]:(3)$$C_{D}=\frac{2k_{1}\mu^{2}d_{f}}{\rho vL\ln k_{2}d_{b}},$$

The shape and orientation factors of the fiber are *k*_1_ and *k*_2_, respectively; *μ* indicates the dynamic viscosity of the gas, *ρ*; the orientation factor, *v*, is the density and velocity of the airflow, respectively; *L* is the characteristic length of EAFMs; and *d_b_* represents the distance between adjacent fibers [142,143,144,145,146].

It can be seen that the size of the pressure drop is also inseparable from the structural factors of the fiber, and the diameter, spacing, and thickness of the fiber affect the size of the pressure drop.

(3) Quality factor

By comparing the overall filtration efficiency (*η*) and the pressure drop (ΔP) of the filtration efficiency, it can be found that the two indexes are contradictory. The influence of structural factors on the pressure drop and filtration efficiency is different. Increased membrane thickness, for example, increases the pressure drop and improves the filtering efficiency. As a result, a quality factor (*QF*) was established to assess the overall performance of the nanofiber membrane. The mathematical model of *QF* [13,145,146,147] is as follows:(4)$$QF=\frac{\ln(1-\eta )}{\Delta P},$$
where *η* is the filtration efficiency (%) and ΔP is the filtration resistance (Pa).

The *QF* evaluates the filtration efficiency of the fiber membrane by combining two factors, avoiding only relying on a single characterization, which would cause the actual quality of the fiber membrane to be unclear. To improve the performance of filtration membranes, researchers can start with these two factors and improve another influencing factor by controlling a single variable to be constant. This approach can effectively improve the filtration performance of fiber membranes. 

### 4.2. The Fabrication of Various Nanofibers for Filtration

Electrospun air-filtration fiber membranes are an ideal filtration material. In recent years, the research on the application of electrospun technology for air filtration has been deepening. At present, there are countless polymers used for electrospun air-filtration materials, and filtration materials that solely contain polymers and the interaction of multiple substances are constantly being developed. Polymers, inorganic matter, and metal particles all have different interactions. Yang et al. [148] prepared polymer and silver nanoparticles to manufacture nanofiber air filter membranes, which not only had an excellent filtration performance, but also had an excellent antibacterial performance against *Escherichia coli* and Staphylococcus aureus. Cui et al. [42] used electrospinning technology to prepare environment-friendly polyvinyl alcohol (PVA)–tannic acid (TA) composite nanofiber membrane filters. At the same time, hydrogen bonds between the TA and PVA polymers improved the mechanical properties of the fiber membrane. In the following subsections, the amount of polymer involved in the preparation of filtration membranes from electrospinning are classified, and the applications of polymer electrospinning in the field of air filtration for single polymers, multiple polymers, and polymer-doped functional particles, respectively, are summarized.

#### 4.2.1. Single-Polymer Nanofibers

At present, in the research on the preparation of air-filtration membranes by the electrospinning process, it is most common that a certain polymer is used for the filtration of PM after being prepared into fiber membranes. The performance of the filtering membrane prepared by electrospinning from a single polymer is mainly dependent on the performance of the material itself and the characteristics of the nanograde materials. For example, Gao et al. [149] used polyvinyl chloride (PVC) to prepare nanofiber filters and successfully realized the removal of particle pollution under conditions of high humidity, which mainly took advantage of the hydrophobicity of PVC. The PVC filtration membrane prepared by Farhangian et al. [150] using the electrospinning process was 1.022 times more effective than common commercial PVC filters, with a higher filtration efficiency and a lower pressure drop, which was mainly attributed to the nanoscale materials prepared by the electrospinning process. The pure polymer air-filter membranes are summarized below, in Table 2.

Compared with traditional filter membranes, air filters prepared by electrospinning have a greatly improved filtration efficiency and pressure drop performance; for example, the filtration efficiency of the PVOH filter membrane prepared by the electrospinning process was improved by more than two times compared with that of the commercial PVOH filter membrane [162]. The environment in which one lives is often not monistic and often involves multiple needs; for example, coal plants require high-temperature resistance and flame-retardant performance, and chemical plants require moisture resistance and chemical corrosion resistance, etc. Filter membranes made of a single polymer often have a single function and cannot meet the needs of complex environments. Therefore, it is concluded that a single polymer can improve its performance after being prepared by the electrospinning process, but it is still not suitable for the demands of all environments.

#### 4.2.2. Multipolymer Composite Nanofibers

The real demand is driving the advancement of process technology, and in order to meet the demands of application in a more complex environment, a variety of polymer materials can handle the problem synergistically. With the real-life complexity of the environment, the demand for air-filtration materials has gradually increased, and materials that meet the requirements of a variety of properties have been continuously studied and produced. There are many ways to prepare air-filter materials by electrospinning various polymers, including the blending, multilayer electrospinning, and multifluid electrospinning processes, as shown in Figure 7. The research on the preparation of composite materials with various polymers has never stopped. Liu et al. [168] prepared core–sheath fibers of TPP and nylon through a coaxial electrospinning process, combining the flame-retardant properties of TPP and the filtration properties of nylon, and applied them to scenarios that are prone to dust explosions and the need to filter particles. In order to solve problems in a complex scenario that filter membranes prepared by a single polymer cannot solve, the research on multiple polymers has also been constantly developing. In the following, the preparation of air-filter materials by the electrospinning process with multiple polymers is summarized and discussed, as shown in Table 3.

The fiber films prepared by the electrospinning of multiple polymers make up for the defect of the singular function of a single polymer to a certain extent, as shown in Table 2. For example, the fiber membrane of lead zirconate titanate/polyvinylidene fluoride (PZT/PVDF) not only has an excellent filtering ability, but also realizes the sensing function of volatile organic compounds (VOCs) [164], and the composite prepared by electrospinning polytetrafluoroethylene–polyamideimide/polyimide (PTEF–PAI/PI) can withstand high temperatures of up to 500 °C and the filtration performance can be adjusted to meet various demands [176]. However, the filter materials prepared by a variety of polymers still cannot meet the growing needs, and the materials still need to be modified, such as by adding antibacterial silver nanoparticles and adsorptive activated carbon.

#### 4.2.3. Nanoparticle-Doped Hybrid Nanofibers

By adding a trace number of functional particles into the materials, the function of the materials is upgraded. Thus, the materials have the capability of meeting different environmental needs. This rule is also followed in the study of preparing air-filter materials by electrospinning. For example, Topuz et al. [177] prepared microporous polyimide metal–organic framework (MOF) nanofiber air-filtration membranes, which greatly improved the trapping ability of VOCs after doping with MOFs. Blosi et al. [178] prepared a filter material of polyvinyl alcohol (PVA) nanofibers doped with trace Ag NPs through an electrospinning process, so that the filter had antibacterial and sterilization abilities on top of the premise of a high filtration efficiency and low air resistance; obviously, the antibacterial performance was due to the addition of the Ag NPs. Materials are modified by adding different functional particles. Common functional particles include Ag NPs, activated carbon, TiO_2_, ZnO, cinnamon essential oil (CO), titanium nanotubes (TNT), and SiO_2_ NPs. See Table 4 for details.

The form of these functional particles doped in polymers is not unique, and the common forms are embedded and dispersed in the fibers, as shown in Figure 8a. Fe_3_O_4_ NPs are uniformly dispersed in the fibers to form uniform and smooth fibers, as shown in Figure 8b, the functional particles are embedded in the surface of the fibers. Through the modification of functional particles, the functions of electrospun fiber membranes become more diversified, such as by adding Ag to increase the antibacterial performance, adding AC to improve the adsorption of VOCs, and adding SiO_2_ to improve the mechanical properties.

Polymers have strong plasticity, and nanofibers prepared by electrospinning technology not only have excellent functionality, but are also easier to modify. The synergy of various polymers and the modification of functional particles mean that electrospinning technology shows great potential in the field of air filtration. It should be noted that the addition of functional particles, besides the diversification of functions, has a great influence on the process parameters of the electrospinning process [195].

#### 4.2.4. Multiprocess Composite Preparation

The three nanofibers discussed above are all prepared by the one-step electrospinning process. As shown in Figure 9, a single polymer is used to prepare nanofibers by electrospinning, which mainly display the inherent characteristics of the nanomaterials and the characteristics of the materials, such as a large specific surface area and small diameter. Multipolymer composite nanofibers combine the properties of various materials to improve the shortcomings of a single material. For example, they can improve the mechanical strength of a material or increase the hydrophilicity of a material. Doping polymer nanoparticles makes the material more functional; for example, adding Ag NPs can make the polymer material have antibacterial properties.

In addition to the preparation of a single process, it is also the focus of research to combine various processes to prepare the required materials. For example, Fan et al. [196] prepared an Ag/PT substrate by the in-situ reduction method, producing an electrospun nanofiber membrane in an anisotropic electric field. Stretched and aligned zein fibers were achieved with simultaneous antimicrobial properties, with a ZNF–Ag/PT filter removing up to 99.30% of the PM0.3, as shown in Figure 10a. This was an effective method to realize a multifunctional air filter by combining various methods to achieve the diversification of functions. Cross-linking materials is a common method to improve the mechanical properties of materials. For example, Cui et al. [182] cross-linked PVA with sodium lignosulfonate by a green chemical cross-linking method, effectively improving the mechanical properties of composite nanofibers and still maintaining a high filtration efficiency of 99.38% after 10 cycles. The combination of various electrospinning processes is a way to prepare functional materials. Park et al. [197] sprayed silver nanowires (AgNWs) on the surface of PAN nanofibers by electrospinning and electrospray. Composite nanofibers have an excellent inhibition ability against all microorganisms. The effect achieved by this method is similar to that of doping nanoparticles, but its realization mechanism is simpler. The requirements for compatibility between materials and particles are lower. Post-treatment processing is one means of material modification. Zhang et al. [198] successfully demonstrated that nanofiber membranes had an excellent antibacterial effect on both Gram-positive bacteria and Gram-negative bacteria through electrospinning preparation and post-treatment chlorination, as shown in Figure 10d. The combination of the synthesis process and the preparation process is not uncommon, and through a specific synthesis process, the material can be given special properties. Then, the material is prepared by the electrospinning process to achieve the result of customizing functional materials. For example, Moon et al. [199] synthesized an amphiphilic PVDF-g-POEM double comb copolymer by ATRP. The nanofiber membrane was further prepared by the electrospinning process, and based on the amphiphilic property, the physical and chemical adsorption capacity of the composite fiber membrane was enhanced.

### 4.3. The Potential of Multifluid Electrospinning Process in Air Filtration

Searching for the topic of “Core sheath nanofiber air filter” in “Web of science” produced almost no results, and searching for the topic of “Janus nanofiber air filter” also yielded nothing. Derivative technology representing electrospinning has not gained a place in the field of air filtration. The core–sheath structure and Janus structure studied by researchers are currently mostly multilayer structures from the macroscopic point of view. For instance, the Janus structure prepared by Xu et al. [200] is visible on the front and back sides of the macroscopic view, but the nanofiber membrane does not have a Janus structure on the microstructure level, and this can be understood as a two-layer filter membrane. The advantages of the coaxial electrospinning process and parallel electrospinning process are that two or more polymers are organically unified without increasing the thickness of the fiber film; for example, core–sheath fiber membranes prepared by coaxial electrospinning can be designed with the core layer as the quality layer and the sheath layer as the functional adsorption layer, and Janus fibers prepared by electrospinning in parallel can combine two materials with different characteristics that contact the air at the same time to form a synergistic effect. Janus nanofibers made by side-by-side electrospinning have the distinct benefit that when air is filtered, pollutant-laden gas must come into close contact with a filtering membrane. Due to the special structure of Janus nanofibers, both materials with different properties are in contact with the outside and their properties are fully utilized. For example, the combination of the piezoelectricity of the PLLA fiber material and the hydrophobicity of PVDF can improve the active adsorption capacity and service life of fiber membranes. Materials prepared by different processes have different characteristics and can meet the needs of the air-filtration process, and the advantages and disadvantages of their air-filtration performance are inextricably linked to the operation mechanism. The research and development of green materials is the need of the times, and the research on degradable materials in other fields is relatively mature. For example, Zhong et al. [201] used CP/PEO nanofiber mats to demonstrate good suitability for fluid filtration and separation. Topuz et al. [202] used renewable cyclodextrin to extract organic raw materials and prepared a strong nanofiber adsorption membrane by electrospinning, which showed good results in water pollution treatment. Green materials also have good application potential in the field of air filtration, which is one of the development directions of materials in the future.

## 5. Conclusions and Outlook

The nanofibers prepared by the electrospinning process have many advantages, and the prepared fiber membranes meet various requirements of an air-filtration material, such as a high porosity, high specific surface area, good pore channel connectivity, and small diameter. These properties can meet the requirements of air filtration and realize the preparation of filter materials with a high filtration efficiency and low air resistance. At the same time, with the development of the electrostatic spinning process, the preparation technology of nanofiber membranes with complex structures has also become mature; for example, coaxial electrospinning, side-by-side electrospinning, and multistage electrospinning can prepare nanofibers with core–sheath structures, Janus structures, and multistage structures. At the same time, research into the use of other functional materials to modify the filter materials is also ongoing. In the near future, the use of more complex nanostructured fibrous materials in the field of air filtration is also foreseeable. The application of electrospinning technology in the field of air filtration is our first step. More in-depth research into the application of complex-structured nanofibers in air filtration is about to be carried out. In the future, it is necessary to improve the research into the multifluid electrospinning process for air filtration and provide more possibilities for fiber air-filter materials.

## Figures and Tables

**Figure 1 nanomaterials-12-01077-f001:**
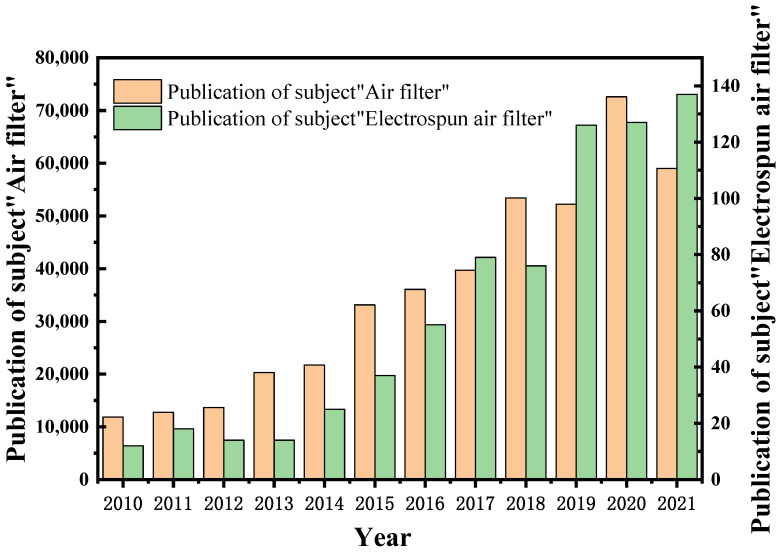
Literature search statistics using the themes “air filter” and “electric spinning and electrospinning air filter” on the “Web of Science” platform.

**Figure 2 nanomaterials-12-01077-f002:**
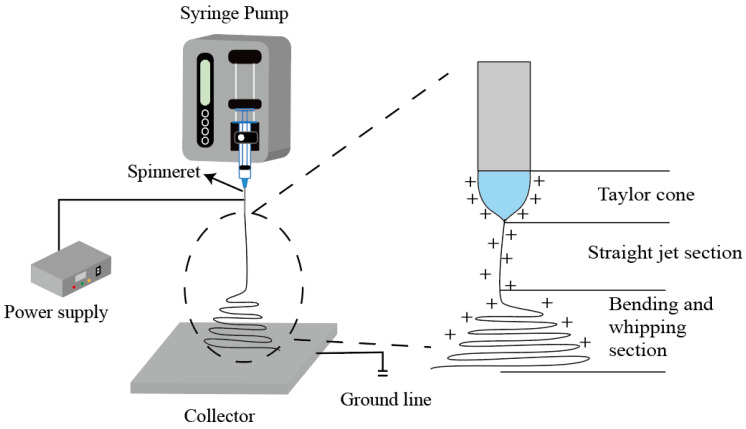
Schematic diagram of electrospinning process and equipment.

**Figure 3 nanomaterials-12-01077-f003:**
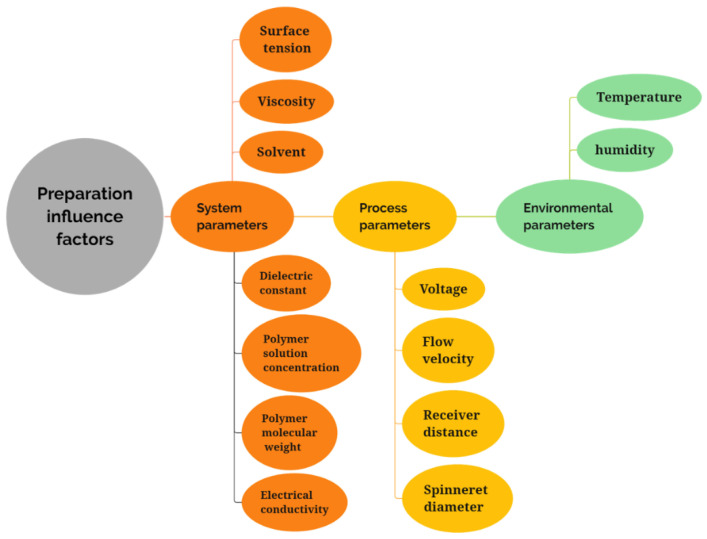
Influencing factors, environmental parameters, process parameters, and environmental parameters of the electrospinning process.

**Figure 4 nanomaterials-12-01077-f004:**
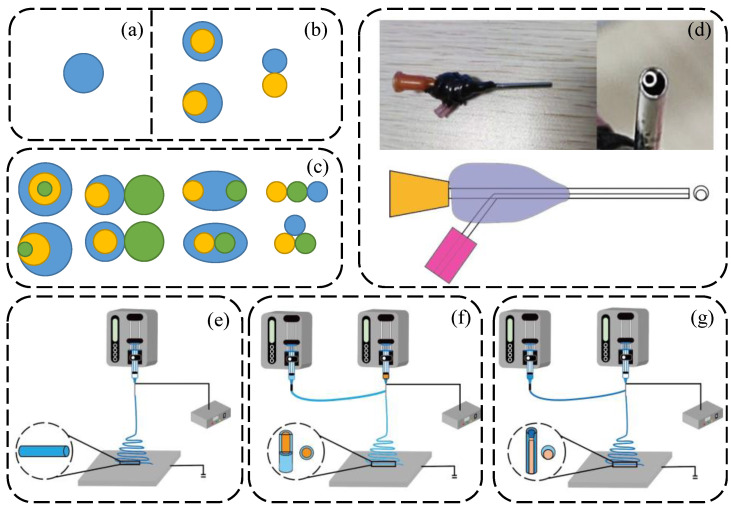
Comparison of electrospinning with different processes: (**a**) single-fluid spinneret; (**b**) two-fluid spinneret; (**c**) three-fluid spinneret; (**d**) real-life photograph and schematic diagram of the eccentric spinning head; (**e**) single-fluid electrospinning schematic diagram; (**f**) coaxial electrospinning diagram; (**g**) parallel electrospinning diagram.

**Figure 5 nanomaterials-12-01077-f005:**
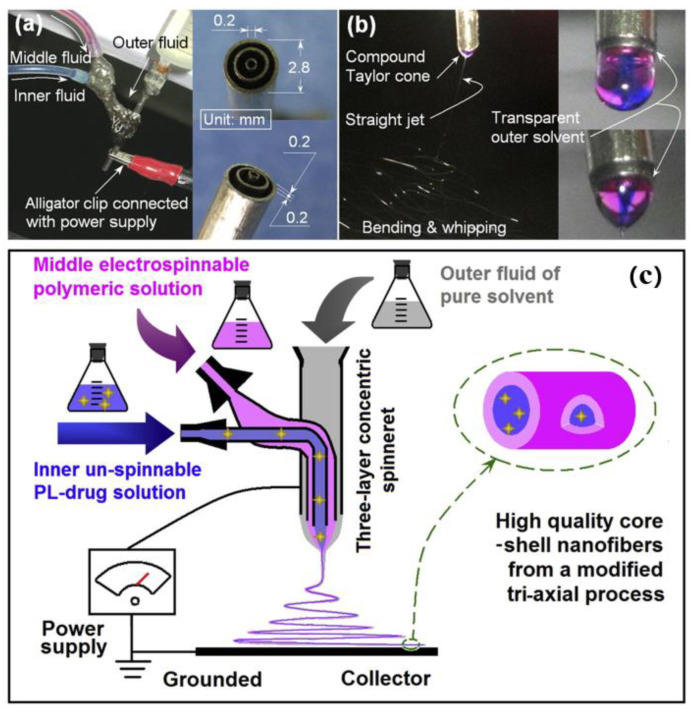
Modified triaxial electrospinning. Adapted with permission from [103]. Copyright, 2016 Elsevier. (**a**) spinneret connections to power and working fluid (**left**), and spinneret image (inset); (**b**) digital photograph of the triaxial process (**left**), droplet (**upper right**), and compound Taylor cone (**lower right**) before application of 15 kV voltage; (**c**) modified triaxial electrostatic spinning process and its use in the preparation of core–shell drug-loaded nanofibers.

**Figure 6 nanomaterials-12-01077-f006:**
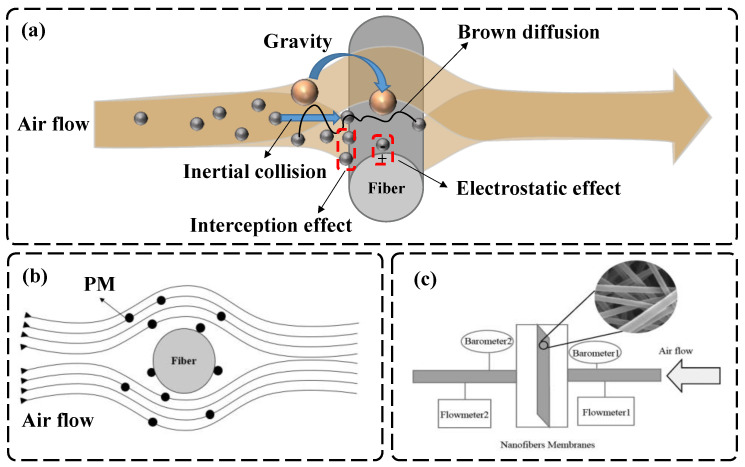
Five mechanisms of fiber filtration: (**a**) filtration mechanisms in the air filtration process; (**b**) aerosol particle motion flowlines; (**c**) filtration test diagram.

**Figure 7 nanomaterials-12-01077-f007:**
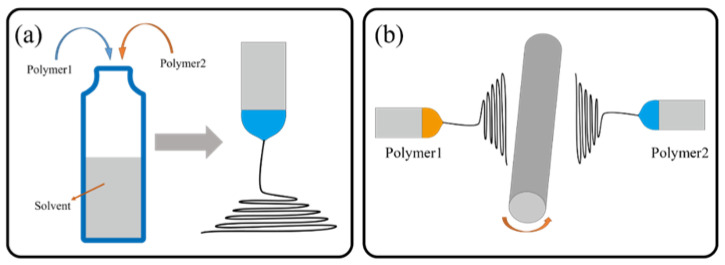
Preparation of air-filter membrane by electrospinning of various polymers: (**a**) blending of various polymers; (**b**) preparing a filter membrane by multiple jets.

**Figure 8 nanomaterials-12-01077-f008:**
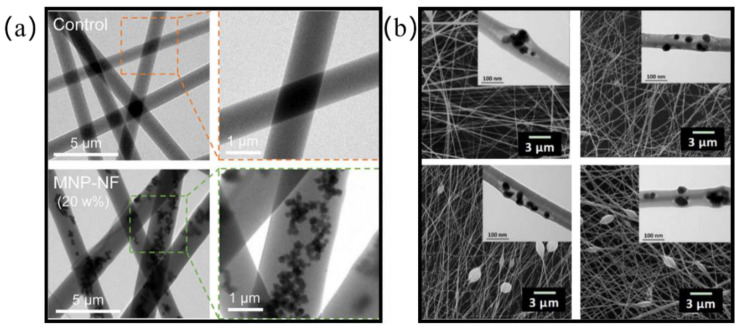
Existing forms of functional particles in fibers: (**a**) Fe_3_O_4_ is uniformly dispersed in fibers Adapted with permission from [179]. Copyright, 2017 American Chemical Society; (**b**) SiO_2_ particles are embedded on the surface of fibers Adapted with permission from [11]. Copyright, 2019 Elsevier.

**Figure 9 nanomaterials-12-01077-f009:**
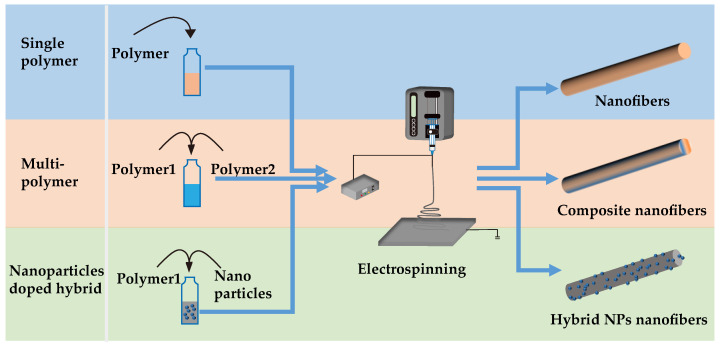
Comparison of preparation processes of single-polymer nanofibers, multiple-polymer composite nanofibers, and nanoparticle-doped hybrid nanofibers.

**Figure 10 nanomaterials-12-01077-f010:**
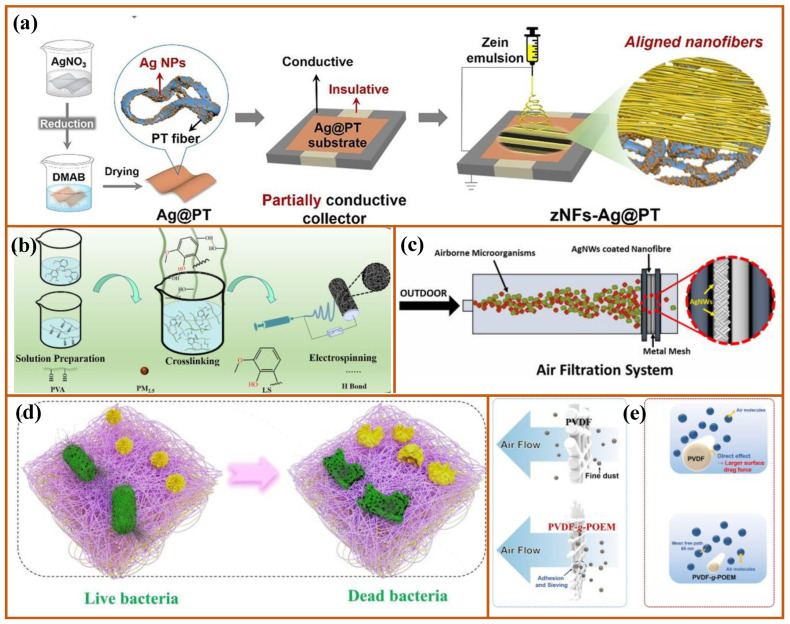
Multiprocess collaborative preparations: (**a**) preparation of Ag/PT substrate by in situ reduction method. Adapted with permission from [196]. Copyright, 2021 American Chemical Society; (**b**) preparation of composite nanofibers by green cross-linking method. Adapted with permission from [182]. Copyright, 2021 Elsevier; (**c**) two-step process of electrostatic spinning and electrospray. Adapted with permission from [197]. Copyright, 2021 Elsevier; (**d**) antibacterial fiber membrane. Adapted with permission from [198]. Copyright, 2020 Elsevier; (**e**) comparison of adsorption efficiency. Adapted with permission from [199]. Copyright, 2021 Elsevier.

**Table 1 nanomaterials-12-01077-t001:** Classification of air-filter materials.

Type	Category	Examples	Advantages	Disadvantages	Refs.
**Traditional filter materials**	Granular filter materials	Coal ash, activated carbon, diatomite, volcanic rock	1. High-temperature resistance 2. Chemical corrosion resistance 3. Low cost	1. Low filtration efficiency 2. High air resistance 3. Serious deficiencies in the filtration of fine particles with particle size <5 μm	[40,41]
	Porous-membrane filter materials	CS/PVA PVDF/PEG PES PVDF/SiO_2_ PAN/F127	1. High filtration efficiency 2. Small aperture	1. Low porosity 2. Poor hole connectivity 3. High air resistance 4. High energy consumption	[42,43,44,45]
	Micron-grade filter materials	Ultrafine glass fiber air-filter materials	1. High-temperature resistance 2. Corrosion resistance 3. Mechanical wear resistance 4. High filtration efficiency	1. Smooth and brittle surface 2. Difficult to process 3. Low bonding strength with the base materials	[46,47]
		Melt-blown electret micron-fiber air-filter materials (PLA, PP, PE)	1. Charge storage capacity 2. High filtration efficiency	1. Limited and easily decayed charge storage capacity 2. Filtration efficiency greatly affected by the charge storage	[48,49]
**Modern filter materials**	Electrospinning nanofiber filter materials	PAN, PCL, PP, PVDF	1. Small aperture 2. Uniform fiber diameter 3. Good tunnel connectivity 4. High porosity 5. High filtration efficiency 6. Small pressure drop 7. Simple process	1. Low production capacity 2. Industrial production has not been fully achieved	[50,51]
	Functional filter materials	PVDF/AgNPs PP/PVA/ZIF-8 PI/ZIF-8	1. High filtration efficiency 2. Strong addition ability	1. Industrial production is difficult 2. High cost	[52]

**Table 2 nanomaterials-12-01077-t002:** Common pure polymer air-filter membranes prepared by electrospinning.

Polymer	Solvents	Special Features	Characteristics	Refs.
PAN	DMF	/	1. Under the airflow rate of 4.2 cm/s, the low pressure drop is 27 Pa, and the efficiency of the PAN filter can easily reach over 99% 2. Realizes a HEPA filter with low bulk density	[14,54,151,152]
		Carbonization modification	1. Excellent thermal stability to temperatures up to 450 °C, supporting the development of autoclaved and recyclable membranes 2. Adsorbs volatile and biological pollutants	[153]
		Surface modification (functional group)	1. PM2.5 removal efficiency of 94.02%, pressure drop of 18 Pa 2. Adsorption performance is enhanced	[154]
PVDF	DMF	/	1. The filtration efficiency is stable for 98.137% to 96.36% 2. The electrospun PVDF fiber also has a certain hydrophobicity, which can effectively improve the service life of the composite filter 3. Charge storage ability and piezoelectricity	[155,156]
PVC	DMF:THF = 1:1	Beaded structure	1. Hydrophobicity 2. The structure improves the filtration efficiency	[149]
		/	1. Filtration efficiency is 1.022 times higher than that of a commercial PVC filter membrane 2. Lower pressure drop	[150]
Nylon	88% formic acid	/	1. High stability 2. The filtration efficiency reaches 99%	[12,157,158]
PA6	99% formic acid	/	1. The dipole moment of the repeating unit in nylon-6 is 3.67 2. The filtering efficiency of the filter is significantly increased to 200% of the original	[159,160]
PA66	99% formic acid	/	1. Good light transmittance 2. The filtering efficiency of PM0.3 reaches 99.99%	[161]
Polyvinyl alcohol (PVOH)	H_2_O	/	1. The performance of repeated use is strong, and it is still good after three uses 2. The filtration efficiency of electrospun PVOH is obviously higher than the original filtration efficiency (3.4 times)	[162]
PLA	DCM/DMF	/	1. The electrostatic charge generated by PLLA nanofibers can significantly improve the application of air filters 2. The efficiency of the PLLA fiber filter is as high as 99.3% 3. Piezoelectricity and degradability	[163]
EC	DMAc/THF	Triboelectricity	1. Friction generates static electricity, which enhances the filtering ability 2. The filtration efficiency reaches 100%	[164]
CA	Ac:DMAc = 3:1	/	1. The electrospinning filter shows good performance at lower thickness 2. Low thickness	[165]
PMMA	DMF	/	1. Strong charge stability 2. The electrospinning film has longer charge retention time and stronger filtration performance stability	[166]
PET	TFA/DCM	/	1. Good mechanical performance, up to 4 MPa 2. High porosity (96%), high collection efficiency (about 100%), and low pressure drop (212 Pa)	[167]

**Table 3 nanomaterials-12-01077-t003:** Multipolymer air-filtration membranes prepared by common electrospinning.

Polymers	Solvents	Technique	Characteristics	Refs.
PZT/PVDF	DMF/Ac	Blend	1. SSSAF has a high filtration efficiency for submicron particles 2. The sensing function of VOC is realized	[169]
PVA/TA	H_2_O	Blend	1. The PM1.0 filtration efficiency reaches 99.5%, and the pressure drop is only 35 Pa 2. The existence of an intermolecular hydrogen bond between PVA and TA improves the mechanical properties of nanofiber membrane	[42]
PVDF/PEI or PVDF/PVA	DMF:Ac = 7:3	Dip and dry coating	1. Captures up to 99.9% of coronavirus aerosols and exhibits superior performance over many commercial masks 2. Virus capture capability	[170]
PAN/PA6	DMF/formic acid	Sequential electrospinning	1. Prepares a plurality of fiber filter screens with different structures (random structure, arranged structure, orthogonal structure, and nanofiber network) by electrospinning 2. Filtration efficiency is improved to 59.46%	[171]
PAN/CTAB	DMF	Blend	1. Removal efficiency of PM2.5 reaches 99.9% 2. The pressure drop is only 11 Pa 3. The quality factor reaches 0.469 PA1	[172]
β-CD/PVA	H_2_O	Blend	1. The filtration efficiency is high (about 99%) and the air permeability is good (the pressure drop is only 45 Pa) 2. Low pressure drop after long-term use	[173]
SF/PVA	H_2_O	Blend	1. SF effectively improves surface properties 2. The filtering effect reaches 99.11%, which is improved by about 10% compared with that of a pure PVA filter core 3. Air drag coefficient reduced to 90% compared with commercial air filters	[174]
PAN/PVP	DMF	Blend	1. The removal rate of fine particles can reach 96.8% 2. Pore structure is improved by heat treatment	[175]
PTEF–PAI/PI	DMAc/H_2_O	Blend/compound	1. Withstands high temperatures up to 500 °C 2. Filtering behavior can be adjusted to meet the requirements of various applications	[176]

**Table 4 nanomaterials-12-01077-t004:** Electrospun filter materials doped with functional particles.

Polymer	Functional Particles	Solvents	Characteristics	Refs.
PVP	Fe_3_O_4_	Ethanol	1. Efficient removal of metal oxide dusts 2. The filtration efficiency reaches 97% and the pressure drop is only 17 Pa	[179]
PVP	Nanoclay	95% ethanol	1. Increase in conductivity 2. Combination of coarse and fine fibers	[180]
CA	AC/TiO_2_	AC/ethanol/2-propanol	1. The pressure drop of ultra-thin filter membrane is 63.0~63.8 Pa 2. CA can be degraded 3. Compared with the undoped filter, the adsorption of charged particles reaches 82% (0.3–0.5 μm)	[181]
PVA	Sodium lignosulfonate (LS)	H_2_O	1. After 10 rounds of circulating filtration, the good air filtration performance is still maintained 2. PVA–LS composite nanofiber membrane shows excellent mechanical properties and transparency	[182]
PVA	Ag NPs	H_2_O	1. Strong antibacterial performance 2. Compared with ordinary commercial products, the filtration efficiency is enhanced	[178]
PVA	Cellulose nanocrystals (CNCs)	H_2_O	1. PVA and CNCs are nontoxic and biodegradable 2. The mechanical strength and the surface charge density of electrospinning solution are increased by CNC doping and thus the fiber diameter is decreased	[183]
PAN	TiO_2_/ZnO/Ag NPs	DMF	1. The diameter of obtained fiber is the smallest by doping TiO_2_, and the filtration efficiency is close to 100% 2. Ag NP-doped fiber has good antibacterial performance	[148,184]
PAN	CNCs	DMF	1. The mechanical properties are improved by two times 2. High filtration efficiency	[185]
PU	AC/CO	DMF:THF = 1:1	1. The nanofiber mat has antibacterial activity 2. AC–CO–PU nanofiber air-filter media can be used in antibacterial fibers, personal masks, air purification devices, and other fields	[186]
PU	Ag NPs/AC	/	1. The adsorption efficiency of volatile gas (VOC) is very high 2. Has an antibacterial effect	[187]
PU	CNCs	DMF	1. The tensile strength and elongation at break increase 2. The filtration efficiency is 99.77%	[188]
PVDF	Titanium nanotubes (TNTs)	MeOH/DMF	1. The bacterial filtration efficiency of 15 wt% TNT/PVDF is 99.88%, which provides greater application potential for clean air management	[189]
PVDF	LiCl	DMF	1. Realization of self-crimping and in situ charging of nanofibers 2. Excellent filtration efficiency (PM0.3 > 99.995%) and low air resistance (55 Pa)	[190]
PVDF	SiO_2_ NPs	DMF	1. High filtration performance 2. High mechanical strength 3. Reusability	[191]
PI	SiO_2_ NPs	DMF/DMAc	1. The tensile strength of PI pure film is increased by 33%, and the tensile strength after solvent steam treatment is increased by 70% 2. High particle (0.3–1.0 µm) filtration efficiency (close to 100%) and stable pressure drop across 20 filtration cycles	[192]
PA6	Ag NPs	Acid	1. The filtration efficiency of PM 2.5 is as high as 99.99%, and the pressure drop is 31 Pa 2. Simultaneously removes various aerosol pollutants, such as SOx, NOx, toluene, and l-nicotine 3. Excellent antibacterial performance against *Escherichia coli* (Gram-negative bacteria) and Staphylococcus aureus (Gram-positive bacteria)	[193]
PU	ZIF-15	DMF/methanol	1. High filtering efficiency of PM2.5 2. Excellent heavy-metal adsorption performance 3. Excellent mechanical properties	[194]

## Data Availability

Not applicable.

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
