# Peer review of "Electrospun Nanofiber Membranes for Air Filtration: A Review"

_nanomaterials, 2022, doi:10.3390/nano12071077_

Round 1

Reviewer 1 Report

Present review is on the fiber prepared by electrospinning method.

It is well organized and will be a good reference for those interested in the field.

One thing I would suggest is to add the section explaining heat treatment process after

electrospinning, which is critical and important to make electrospun fibers stable

under various conditions.

Author Response

On behalf of all the co-authors, we highly appreciate the reviewer for the precious time and efforts spent on our manuscript, and the pertinent comments and suggestions as well. Accordingly, we have tried our best to improve the manuscript. Our point-by-point responses are listed as follows:

Present review is on the fiber prepared by electrospinning method. It is well organized and will be a good reference for those interested in the field. One thing I would suggest is to add the section explaining heat treatment process after electrospinning, which is critical and important to make electrospun fibers stable under various conditions.

Response: Many thanks for your comments, as you have stated, the effect of heat treatment on nanofibers after electrospinning is powerful. This approach is not only effective in enhancing the stability and mechanical properties of nanofibers, but also in modulating the internal structure and achieving small size effect. These advantages make the application of electrospun fibers in air filtration more promising. We have added this section to the article as follows:

“Besides the properties of raw materials and the influencing factors in the prepara-tion process, the post-treatment of products is also an important factor affecting the performance. For example, heat treatment of products can improve the mechanical properties and stability of materials. In the face of the industrial waste gas emission as high as 260℃, it is urgent to prepare refined and high-temperature resistant filter ma-terials. The electrospun fiber prepared at room temperature is easy to be oxidized at high temperature, and the filtration efficiency is reduced. By designing heat treatment conditions, the structure of electrospun fiber membrane was improved to achieve the advantages of small size effect, adjusting internal connectivity, improving stability and enhancing mechanical properties. Chen et al. [78] prepared electrospun nanofibers by blending PA66 and PVB. Through SEM images, it is obvious that the fiber diameter decreases and the regular arrangement of more regular fiber structures. Compared with untreated nanofibers, the filtration efficiency of W-15G nanofiber membrane for 0.3μm airborne particles is as high as 99%. The cross-link agent is added in that com-posite fib, and the rapid cross-linking of the composite fiber is realized through the heat treatment process. This method can significantly improve the solvent resistance and mechanical properties of nanofibers. Gui et al. [79] annealed at 170℃ by adding a certain amount of citric acid into the working solution of electrospinning precursor. When the annealing temperature is 160℃, the tensile strength of CS/PEO/CA nanofiber felt is as high as 8.11Mpa. In addition, for the nanoparticle-nanofiber mixture, heat treatment can not only reduce the fiber diameter but also improve the crystallinity of nanocrystals. Zhang et al. [80] prepared composite fiber membrane by electrospinning water-soluble PVA/CNCs. The existence of CNCs significantly promotes the growth of PVA crystals under heat treatment. Zhang et al. [81] heat-treated PAN/PMIA compo-site fiber membrane at 140~220℃ for 30min. There is no entanglement between the fibers, and the breaking strength of the composite fiber film decreases slightly, while the elongation at break increases significantly. After heat treatment, the filtration effi-ciency of PAN/PMIA composite fiber membrane remains above 99%. In summary, heat treatment can improve the performance of nanofibers after preparation, such as fiber stability and mechanical properties, as one of the post-treatment methods for materials. Therefore, reasonable post-treatment of products is also one of the ways to improve materials properties.”

Please see 3.2.4 part in page 9.

Once more, many thanks to you for the precious time and efforts spent on our manuscript. As a result of the valuable comments and suggestions, we believe we have been able to improve the quality of our manuscript. We sincerely hope our manuscript is now suitable for publication in Nanomaterials and look forward to hearing from you at your earliest convenience.  

Best regards,

Dr. Ke Wang

Reviewer 2 Report

  1. Avoid using abbreviations in the title as the journal has a very broad readership and might not be obvious for all readers what PM means.
  2. The synthesis of CTAB-functionalized large-scale nanofibers air filter media for efficient PM2.5 capture capacity with low airflow resistance should be discussed (10.1021/acsapm.0c01203).
  3. Some comparative figures on the different fabrication methodologies as well as applications should be added in the manuscript.
  4. Nanofibrous membranes comprising intrinsically microporous polyimides with embedded metal–organic frameworks for capturing volatile organic compounds should be mentioned (10.1016/j.jhazmat.2021.127347).
  5. Electrospun polyurethane/zeolitic imidazolate framework nanofibrous membrane with superior stability for filtering performance is missing from the review (10.1021/acsapm.0c01005).
  6. Discuss electrospun membranes filtering 100 nm particles from air flow by means of the van der Waals and Coulomb forces (10.1016/j.memsci.2021.120138).
  7. Sustainable and green approaches for electrospun nanofiber membranes are emerging and important aspects that should be widely applied. The review should mention some of these efforts (10.1016/j.carbpol.2021.118593; 10.1016/j.cej.2021.129443) and the need to extend to air filtration materials as well.
  8. Electrospun polystyrene and acid-treated cellulose nanocrystals with intense pulsed light treatment for N95-equivalent filters, and cellulose nanocrystal filters need to be acknowledged (10.1021/acsapm.1c00722; 10.1016/j.memsci.2022.120392).
  9. The influence of electrospinning parameters and salt addition is important and should be briefly mentioned (10.1016/j.matdes.2020.109280).
  10. Durable superhydrophobic poly(vinylidene fluoride) (PVDF)-based nanofibrous membranes for reusable air filters should be mentioned (10.1021/acsapm.1c01314).

Author Response

On behalf of all the co-authors, we highly appreciate the reviewer for the precious time and efforts spent on our manuscript, and the pertinent comments and suggestions as well. Accordingly, we have tried our best to improve the manuscript. Our point-by-point responses are listed as follows: 

1.    Avoid using abbreviations in the title as the journal has a very broad readership and might not be obvious for all readers what PM means.
Response: Yes, many thanks. We changed' PM' to 'Air' to enhance the readability of the title, and at the same time, make the article and the title more closely related. Please see the change in title.

2.    The synthesis of CTAB-functionalized large-scale nanofibers air filter media for efficient PM2.5 capture capacity with low airflow resistance should be discussed (10.1021/acsapm.0c01203).
Response: Yes, Thank you very much for your suggestion. We added some discussion in Table 3, we cite this paper in our revised manuscript as Ref.[172], which perfectly improved the content of the table. Please see the changes in Table 3 “PAN/CTAB” section in page 20.

3.    Some comparative figures on the different fabrication methodologies as well as applications should be added in the manuscript.
Response: Many thanks to your kind comments. We have added the comparison of different preparation methods and applications, as shown in the Figures as following. Firstly, we compare the preparation methods and characteristics of single polymer nanofibers, multiple-polymer composite nanofibers and nanoparticles doped hybrid nanofibers. As shown in Figure 9, the preparing process of the various nanofibers from electrospinning solution and the type of the product are performed. Briefly explain the main performance sources and characteristics of these three materials. 

Figure 9. Comparison of preparation processes of single polymer nanofibers, multiple-polymer composite nanofibers and nanoparticles doped hybrid nanofibers.
Then we describe the composite preparation of nanofiber filter materials by various processes. From the combination of in-situ synthesis and electrospinning, the combination of green cross-linking method and electrospinning, the combination of various electrospinning processes, electrospinning and post-treatment of products are illustrated with examples. As shown in Figure 10. The detailed discussion and corresponding Figure we have added in the part of 4.2.4, please see the page 24-25.

Figure 10. Multi-process collaborative preparation (a) preparation of Ag@PT substrate by in-situ reduction method [196]; (b) preparation of composite nanofibers by green cross-linking method [182]; (c) two-step process of electrostatic spinning and electrospray [197]; (d) Antibacterial fiber membrane [198]; (e) Comparison of adsorption efficiency [199].

4.    Nanofibrous membranes comprising intrinsically microporous polyimides with embedded metal–organic frameworks for capturing volatile organic compounds should be mentioned (10.1016/j.jhazmat.2021.127347).
Response: Many thanks for recommending us those excellent articles, correspondingly, we are pleased to cite it in our revised manuscript as Ref. [177]. As you suggested, the addition of MOFs can obviously improve the adsorption performance of volatile organic compounds (VOCs). We have elaborated on this part in more detail to complement its deficiencies, please see 4.2.3 part in page 21.

5.    Electrospun polyurethane/zeolitic imidazolate framework nanofibrous membrane with superior stability for filtering performance is missing from the review (10.1021/acsapm.0c01005).
Response: Thank you very much for supplementing this part. We have supplemented the excellent performance of polyurethane/zeolite imidazole skeleton nanofiber membrane; we are pleased to cite it in our revised manuscript as Ref. [194]. Please see Table 4 part in page 23.

6.    Discuss electrospun membranes filtering 100 nm particles from air flow by means of the van der Waals and Coulomb forces (10.1016/j.memsci.2021.120138).
Response: Yes,many thanks. Van der Waals and coulomb force have a remarkable influence on tiny particles, which has been added to the article. We cite it in our revised manuscript as Ref. [137]. Please see 4.1.1 part in page 15.

7.    Sustainable and green approaches for electrospun nanofiber membranes are emerging and important aspects that should be widely applied. The review should mention some of these efforts (10.1016/j.carbpol.2021.118593; 10.1016/j.cej.2021.129443) and the need to extend to air filtration materials as well.
Response: Many thanks to your suggestion. Green materials and renewable energy are the focus of sustainable development and the direction of future development,we are pleased to cite them in our revised manuscript as Ref.[201] and [202]. Please see 4.3 part in page 26.

8.    Electrospun polystyrene and acid-treated cellulose nanocrystals with intense pulsed light treatment for N95-equivalent filters, and cellulose nanocrystal filters need to be acknowledged (10.1021/acsapm.1c00722; 10.1016/j.memsci.2022.120392).
Response: Yes,Thank you very much for reminding us of this missing part. We have supplemented the doping content of CNCs and summarized the relevant features in Table 4, we cite them in our revised manuscript as Ref. [188] and [184], please see page 22.

9.    The influence of electrospinning parameters and salt addition is important and should be briefly mentioned (10.1016/j.matdes.2020.109280).
Response: Thank you very much for reminding me. We have made a brief supplement to the influence caused by adding functional particles, we cite it in our revised manuscript as Ref. [195], please see page 24.

10.    Durable superhydrophobic poly(vinylidene fluoride) (PVDF)-based nanofibrous membranes for reusable air filters should be mentioned (10.1021/acsapm.1c01314).
Response: Thank you very much for your suggestion. PVDF@SiO2NPs fiber membrane has good reusability. We have already supplemented this part, we cite it in our revised manuscript as Ref. [191], please see Table 4 in page 23.

Once more, many thanks to you for the precious time and efforts spent on our manuscript. As a result of the valuable comments and suggestions, we believe we have been able to improve the quality of our manuscript. We sincerely hope our manuscript is now suitable for publication in Nanomaterials and look forward to hearing from you at your earliest convenience.  

Best regards,
Dr. Ke Wang

Round 2

Reviewer 2 Report

The manuscript has been updated according to the comments.